# Correlation between Physical Performance and Tensiomyographic and Myotonometric Parameters in Older Adults

**DOI:** 10.3390/healthcare11152169

**Published:** 2023-07-31

**Authors:** Noé Labata-Lezaun, Vanessa González-Rueda, Luis Llurda-Almuzara, Carlos López-de-Celis, Jacobo Rodríguez-Sanz, Aida Cadellans-Arróniz, Joan Bosch, Albert Pérez-Bellmunt

**Affiliations:** 1Department of Basic Sciences, Faculty of Medicine and Health Sciences, Universitat Internacional de Catalunya, 08195 Sant Cugat del Vallès, Spain; nlabata@uic.es (N.L.-L.); jrodriguezs@uic.es (J.R.-S.); jboschs@uic.es (J.B.); aperez@uic.es (A.P.-B.); 2ACTIUM Functional Anatomy Group, Universitat Internacional de Catalunya, 08195 Sant Cugat del Vallès, Spain; vgonzalez@uic.es (V.G.-R.); luis.llurda@euneiz.com (L.L.-A.); 3Physiotherapy Department, Faculty of Medicine and Health Sciences, Universitat Internacional de Catalunya, 08195 Sant Cugat del Vallès, Spain; acadellans@uic.es; 4Fundació Institut Universitari per a la Recerca a l’Atenció Primària de Salut Jordi Gol i Gurina (IDIAPJGol), 08007 Barcelona, Spain; 5Physiotherapy Department, Faculty of Health Sciences, European University of Gasteiz—EUNEIZ, La Biosfera Ibilbidea, 6, 01013 Vitoria-Gasteiz, Spain

**Keywords:** elderly, physical functional performance, tensiomyography, myotonometry, muscle quality

## Abstract

Background: To examine the correlation between physical performance and muscle strength and the variables obtained from tensiomyography and myotonometry. Methods: Fifty-two older adults able to complete functional tests participated in this observational study. Variables of maximal radial muscle displacement (Dm) and contraction time (Tc) (using tensiomyography) and muscle stiffness (using myotonometry) of the rectus femoris and vastus lateralis muscles were assessed. Physical performance (Short Physical Performance Battery, Timed Up and Go, Five Times Sit to Stand, and walking speed), isometric knee extension strength, and grip strength were assessed. A correlation analysis was performed between all the variables. Results: A significant correlation between the Short Physical Performance Battery and the rectus femoris (rho = 0.491) and vastus lateralis Dm (rho = 0.329) was found. Significant correlations between the Five Times Sit to Stand Test and the Dm values of the rectus femoris (rho = −0.340) and Dm (rho = −0.304), and stiffness (rho = −0.345) in the vastus lateralis, were also found. No significant correlations were found between tensiomyography and myotonometry, the Timed Up and Go, and walking speed, nor between tensiomyography and myotonometry and grip strength or isometric knee extension strength. Conclusions: Functional tests should be prioritized in the assessment of older adults, but further research into muscle quality using technology is advisable.

## 1. Introduction

Recently, the World Health Organization (WHO) has defined healthy aging as “the process of developing and maintaining functional capacity that enables well-being in old age” [1]. In this way, the focus has been placed on the relevance of preserving, as far as possible, the functional status of the older adult in order to prevent, delay, or mitigate the consequences of age-related pathologies.

Skeletal muscle aging involves a decline in muscle mass, strength, and function, leading to mobility issues and increased risks of falls and disability [2]. Therefore, assessing muscle function related to atrophy in older individuals before it occurs is of clinical importance [3]. In that sense, the European Working Group on Sarcopenia in Older People (EWGSOP), in a recent consensus, has included the importance of assessing muscle strength and physical performance as main factors when diagnosing a pathology as prevalent in older adults as sarcopenia [4]. For this purpose, different functional tests such as the grip strength measurement, the sit-to-stand test, or the Short Physical Performance Battery (SPPB) have usually been used [5,6]. For example, grip strength is a widely used marker in both clinical and research settings for assessing muscular strength [6,7]. In fact, it is considered a good biomarker of health [8,9], and low grip strength values predict the onset of conditions such as frailty and sarcopenia, as well as other adverse events such as falls, hospitalization, or death [10,11]. Despite not being a direct measure of lower limb strength, several studies have concluded that there is a correlation between grip strength and knee extension strength [12,13].

In addition, the EWGSOP comments on the importance of assessing not only muscle quantity but also muscle quality (micro and macroscopic aspects of muscle architecture and composition), recognizing the difficulty in establishing a precise definition as well as the lack of devices that are currently capable of assessing it. It is imperative to conduct research on the construct of muscle quality and its alignment with parameters of physical performance and muscular function [4]. New technologies, together with current knowledge of physiology, have made it possible to assess muscle quality. Among these technologies, tensiomyography and myotonometry stand out [14]. Tensiomyography is a valid and reliable assessment method [15] used to evaluate muscle quality by measuring the radial displacement of the transverse fibers of the muscle belly as a function of contraction time. Previous studies have shown that tensiomyography is capable of assessing the proportion of type I fibers in a muscle. Lower Tc values would be related to a lower proportion of slow fibers [16]. In older adult populations, there is an increase in muscle contraction time caused by a decrease in type II fibers [17], which are activated mainly in actions requiring strength and speed [18]. Therefore, indirectly, it could be hypothesized that an increase in Tc due to the loss of type II fibers could be associated with a decline in maximum strength. However, currently, there are no studies available on this matter. On the other hand, tensiomyography has also been able to detect changes in muscle stiffness in a phase of atrophy during a period of immobilization [19]. A single study has studied the relationship between physical performance and tensiomyography in older adults, finding contradictory results [20]. Myotonometry is another reliable non-invasive assessment method used to evaluate the viscoelastic properties of tissues [21], with different parameters from those observed in tensiomyograpghy [22]. There are currently no studies linking myotonometry and physical performance. The lack of studies with both technologies in older adults makes it difficult to establish normative values for this population.

Knowing the changes that occur in muscle quality through the use of new technologies and how these changes are related to physical performance will make it possible to optimize the screening process for older adults. This could allow us to identify those at risk for negative events (risk of falls, frailty, etc.) and be able to establish prevention strategies as well as objectively monitor their evolution. Finally, understanding the factors contributing to muscle aging is crucial to developing effective interventions for healthy aging. To date, it is still unclear whether there is a direct relationship between the deterioration in muscle quality and the decrease in strength and physical performance in the elderly. Thus, the main objective of the present project is to analyze the level of correlation between physical performance and muscle strength tests and the variables obtained from tensiomyography and myotonometry. The main hypothesis of the authors is that there should be a strong correlation between physical performance, muscle strength, and muscle quality parameters.

## 2. Materials and Methods

### 2.1. Study Design

A cross-sectional observational correlational study was conducted in accordance with the Strengthening the Reporting of Observational Studies in Epidemiology (STROBE) Checklist [23]. The study received approval from the Research Ethics Committee of the Universitat Internacional de Catalunya (CBAS-2020-12). Data collection, processing, and transfer were carried out in accordance with the Declaration of Helsinki [24] and the prevailing Spanish legislation regarding the protection of personal data.

### 2.2. Participants

Participants were non-randomly recruited according to their availability to participate. The inclusion criteria for this study were individuals aged 65 and older. Exclusion criteria included: (a) inability to stand or walk without assistance; (b) history of bone fracture within the past 6 months; (c) uncontrolled symptomatic cardiovascular or respiratory conditions; (d) current cancer diagnosis; and (e) inability to comprehend the information provided by the assessors.

### 2.3. Sample Size Calculation

The sample size calculation was performed using G*Power software (version 3.1). For the main variable, SPPB, no previous studies were found, so a moderate correlation level of r = 0.45 was predicted. For the second main physical performance variable, Timed Up and Go (TUG), the sample size was calculated using the data obtained in a similar study by Fabiani et al. [20]. In the study, the authors found a correlation between contraction time (Tc) and TUG of rho = 0.456 in the vastus lateralis quadriceps musculature. Establishing an α error of 0.05 and a statistical power of 0.95 and assuming losses of 10% due to possible errors in data transcription, a sample size of 50 subjects was obtained for the SPPB and 52 for the TUG. Choosing the higher value, the sample size was set at 52 subjects.

### 2.4. Procedure

Once the subjects were contacted, they were confirmed to meet the inclusion/exclusion criteria before the evaluation. First, personal data were recorded, as well as height, weight, and dominance. Next, tensiomyography and myotonometry tests were performed on the rectus femoris and vastus lateralis muscles of the dominant limb. To conduct the tensiomyographic measurements, the patient was positioned in a relaxed muscle state, typically in the supine position. Two electrodes (TMG electrodes, TMG-BMC d.o.o., Ljubljana, Slovenia) were placed on the most prominent area of the selected muscle belly, with a 5 cm distance between the electrodes. Additionally, the Dc-Dc Trans-Tek^®^ transducer (GK40, Panoptik d.o.o., Ljubljana, Slovenia) was positioned perpendicularly. An electrostimulator device (TMG-BMC d.o.o., Ljubljana, Slovenia) connected to the electrodes was used to induce involuntary muscle contractions, resulting in radial displacement of the sensor and generating a time–displacement curve. The amplitude of stimulation was gradually increased from 20 to 100 mA in 20 mA increments until no further increase in Dm (displacement of the muscle) was observed or the maximum capacity of the stimulator was reached (i.e., 110 mA). Ten seconds of rest was allowed between stimuli to minimize the effects of fatigue and post-activation potentiation [25]. To conduct the myotonometric measurements (MytonPro^®^, Myoton Ltd., Tallinn, Estonia), the participant was positioned in a relaxed muscle state. The sensor of the device was placed perpendicularly on the most prominent area of the selected muscle belly, focusing solely on superficial musculature. A gentle pressure was applied to the tissue with the device, allowing for three brief pressure applications (15 ms each) to be performed on the tissue [25]. Isometric knee extension strength was assessed with a hand-held dynamometer. Finally, functional tests (SPPB, TUG, 5XSST, and 4 m walking speed) and grip strength assessments were performed.

### 2.5. Variables

#### 2.5.1. Muscle Quality

Maximal radial muscle displacement (Dm): This tensiomyographic parameter [15,26] indicates the maximum radial displacement (in millimeters) of the muscle when subjected to involuntary contraction by electrostimulation [22]. The rectus femoris and vastus lateralis muscles were measured by taking one measurement for each of the muscle groups.

Contraction time (Tc): This tensiomyographic parameter indicates the time (in milliseconds) it takes for the muscle to reach from 10 to 90% of the maximum radial displacement (Dm) [27]. The rectus femoris and vastus lateralis muscles were measured.

Stiffness: This myotonometric parameter is determined by the ratio between the force produced by the mechanical impulse and the depth of tissue deformation. Its unit of measurement is newtons per meter (N·m^−1^) and is calculated based on the following equation: S = α_max_ · m_probe_/Δl (m = mass of the testing end, α_max_ = maximum deformation acceleration of the tissue, Δl = maximum displacement of the tissue) [28]. The femoris and vastus lateralis muscles were measured.

#### 2.5.2. Physical Performance

Short Physical Performance Battery (SPPB): This is a widely utilized functional test battery in primary care and research settings. It encompasses three tests, including a balance test in three different positions (feet together, semi-tandem, and tandem) for 10 s with eyes open, a 4 m walking test, and a five-chair raise test. Each test yields a numerical value ranging from 0 to 4, which is then summed to obtain a maximum overall score of 12 points (Figure 1A). The SPPB has demonstrated good to excellent test–retest reliability (ICC between 0.83 and 0.92) and excellent inter-rater reliability (ICC 0.91) among elderly patients [6].

Timed Up and Go Test (TUG): The test measures the duration, in seconds, it takes for an individual to rise from a chair with the aid of their arms, walk a distance of 3 m as quickly as possible, navigate around an obstacle, return to the chair, and sit down again (Figure 1B). The test was administered twice, and the shorter time recorded was used for analysis. Previous studies have demonstrated high reliability for this test (ICC = 0.98, 95%CI = 0.93–1.00; SEM = 0.7) [6].

The 4-Meter Walking Test (WT): This test measures the average speed, in meters per second, that an individual achieves during a 4-Meter Walking Test at their usual pace. Although it is part of the SPPB battery, the test score holds independent value. The test was conducted twice, and the time from the trial with the shortest duration was selected. Previous studies have established the reliability of this test (ICC = 0.96, 95%CI = 0.94–0.98; SEM = 0.01) [29].

#### 2.5.3. Muscle Strength

Five Times Sit to Stand Test (5XSST): This test assesses the duration, in seconds, required for an individual to perform five repetitions of sitting down and standing up from a chair with a backrest without utilizing their arms for assistance. Although it is part of the SPPB battery, the test score holds independent value. The test was conducted twice, and the time from the trial with the shortest duration was selected [4].

Handgrip strength (HS): Measured by hand dynamometry. Its unit of measurement is kilograms (Kg) [30]. The Jamar Plus+^®^ (Sammons Preston, Rolyon, Bolingbrook, IL, USA) dynamometer device was used. For the measurements, the participant was seated on a chair with their back resting against the backrest, feet placed on the floor, and arms relaxed, maintaining a 90° elbow flexion and neutral wrist position. Both the dominant and non-dominant arms were measured. Three measurements were taken alternately for each hand, with a one-minute rest between measurements. The average of the three measurements was calculated for each hand, and the hand with the superior results was selected. The validity and reliability of this device have been assessed in prior studies (ICC = 0.98) [31].

Knee extension muscular strength (KE-MS): Measured with hand-held dynamometry. Its unit of measurement is kilograms (kg). The knee extension of the dominant limb is measured. For the procedure, the subjects were placed in a seated position with their knees flexed at 90°. From that position, the participants were asked to perform the knee extension movement, which was resisted with the use of a strap attached to a fixed anchor. The dynamometer was fixed, together with the strap, to the distal third of the tibia. It is a valid and reliable method (ICC = 0.78, 95%CI = 0.63–0.92; SEM = 13.87) [32]. The hand-held dynamometer MicroFET 2^®^ device (Hoggan Scientific LLC., Salt Lake City, UT, USA) was used [33]. During the measurement, a strap was placed to fix the dynamometer to the subject in such a way as to ensure that the contraction was isometric. Three measurements were taken for each movement, and the average of the three was calculated.

### 2.6. Statistical Analysis

Statistical analysis was conducted using Jamovi software version 1.6.23. Descriptive statistics were computed for all variables, with quantitative variables reported as mean and standard deviation and qualitative variables presented as frequencies and percentages. The normal distribution of variables was assessed using the Shapiro–Wilk test. Correlation analysis was performed using either Pearson’s correlation coefficient or Spearman’s rank correlation coefficient, depending on the distribution of the variable being evaluated. A significance level of 0.05 was set, with a 95% confidence interval. The strength of the correlation coefficient was interpreted as follows: 0–0.10 indicating an insignificant correlation, 0.10–0.39 indicating a weak correlation, 0.40–0.69 indicating a moderate correlation, 0.70–0.89 indicating a strong correlation, and 0.90–1.00 indicating a very strong correlation [34].

## 3. Results

The characteristics of the 52 participants are shown in Table 1. The descriptive values of the variables are presented in Table 2. Of the data of the 52 participants, 2 values of the tenisomyography variables were lost due to an error in the transcription of the data. Similarly, 2 values of handgrip, 1 of SPPB, and 1 of TUG were lost. None of these losses exceeded 10% of the losses assumed in the sample calculation.

Table 3 shows the correlation matrix between the different physical performance assessment tests and the tensiomyography and myotonometry variables. The correlation analysis shows a significant relationship between the SPPB test and the values of Dm of the rectus femoris (rho = 0.491, *p* < 0.001) and Dm of the vastus lateralis (rho = 0.329, *p* = 0.021) and the 5XSST test and the values of Dm of the rectus femoris (rho = −0.34, *p* = 0.016), vastus lateralis Dm (rho = −0.304, *p* = 0.032), and vastus lateralis stiffness (rho = −0.345, *p* = 0.012). Correlation diagrams for statistically significant variables are shown in Appendix A.

## 4. Discussion

The aim of this study was to analyze the level of correlation between the functional and muscle strength tests and the variables obtained from tensiomyography and myotonometry.

With respect to the descriptive values, all the variables obtained in the functional tests, as well as the grip strength, were found to be above the cut-off points defined by the EWGSOP for the diagnosis of sarcopenia [4]. This is a population that would be at low risk of suffering future negative events, such as care dependency, falls, fractures, or mortality [6]. Regarding tensiomyography and myotonometry values, there are no normative values for the knee extensor musculature in the older adult population. Thus, the results of the present study allow us to know the values for a healthy population. Fabiani et al., 2021 [20] analyzed, in a sample of 28 women over 65 years of age, the Tc for the vastus lateralis quadriceps muscle, obtaining a slightly higher contraction time. Similarly, regarding the Dm of the vastus lateralis, his results were superior. Teraž et al. [35] conducted an analysis on a sample of 52 older adults. The descriptive results obtained in our study for the vastus lateralis muscle demonstrate lower values in terms of Dm and higher values in terms of Tc as compared to their study. Finally, Jacob et al. found similar values in Dm but higher values in Tc in a sample of 51 older adults [36]. No studies were found that assessed the rectus femoris in this population. Regarding myotonometry, Agyapong-Badu et al. published an article in which they analyzed the stiffness of the rectus femoris in a similar population [37]. In all three studies, the values were higher than those obtained in our sample. Comparing the results with those of a population of young people, rectus femoris stiffness was found to be lower in both a sample of inactive young people [37] and athletes [38]. The stiffness results of the present study were lower than those of a population of master athletes [38]. These results suggest that muscle tissue becomes stiffer with aging. A certain level of muscle stiffness could be beneficial in transmitting muscle contraction to the joints more efficiently. It would be understandable in this case that master athletes would have a higher stiffness than non-athletes. However, studies performed on the tibialis anterior and gastrocnemius anterior musculature show a decrease in stiffness with age [39].

Regarding the correlation analysis, a statistically significant moderate positive correlation was found between the SPPB battery and the Dm of the rectus femoris and vastus lateralis quadriceps musculature. In addition, a statistically significant weak negative correlation was found between the 5XSST test and the Dm of the rectus femoris and vastus lateralis quadriceps musculature. These results would indicate that a better physical performance would be related to higher Dm values. Previous studies on the topic are contradictory. On the one hand, studies conducted by Pišot et al. [40] and Šimunič et al. [19] showed that a 35-day period of bed rest resulted in an increase in Dm. These results would indicate an increase in Dm during a period of disuse muscle atrophy. It should be noted that the participants in both studies were young adults. On the other hand, Paravlić et al. [41] found a decrease in Dm in the knee extensor musculature after one month of undergoing total knee arthroplasty in older adults. Three studies aimed to correlate physical performance and Dm in an older adult population [20,35,36]. In agreement with the results of the present study, they found a negative correlation between both variables. The possible hypothesis proposed by the authors regarding these results was that the infiltration of fat and connective tissue within the muscles produced by aging could influence this correlation. In our opinion, it seems that this hypothesis has not yet been demonstrated; however, in view of the results, it would seem clear to differentiate muscle atrophy produced by disuse from muscle atrophy related to aging.

Regarding myotonometry, a statistically significant weak negative correlation was also found between the 5XSST test and vastus lateralis quadriceps stiffness. This correlation would imply that people with low physical performance would have lower stiffness. No previous studies have been found to corroborate the present results. Only the study by Agyapong-Badu et al. [42] generated a musculoskeletal health status discrimination model based on different functional tests and technologies, including the Myoton.

No statistical significance was found for the rest of the functional tests or for the strength tests. To our knowledge, until now only the article by Fabiani et al. [20] has correlated tensiomyography with physical performance assessment tests (specifically with the TUG test). In addition to the previously discussed Dm values, they found a moderately negative correlation between TUG values and vastus lateralis Tc values. These results would imply that a decrease in contraction time would be related to a lower level of physical performance. It is plausible to conclude from the study of Fabiani et al. [20] that this decrease in the number of type II fibers could be the cause of the decrease in physical performance. In this sense, Zubac et al. [43] performed a plyometric training intervention in older adults, obtaining significant improvements in the countermovement jump take-off speed (CMJ) and a decrease in Tc. Likewise, Šimunič et al. [17] conducted an observational study in which they demonstrated that master athletes in power sports maintained lower Tc values than a group of non-athlete older adults. Unfortunately, the results of the present study show no correlation between contraction time and physical performance, so further studies will be necessary to confirm this hypothesis.

As recommended by the EWGSOP in its latest consensus, it is a priority to identify which indicators of muscle quality best predict the loss of muscle mass, strength, and function. In the same way, it raises the question of how and through which tools muscle quality can be accurately and affordably assessed [4]. In this sense, tensiomyography appears as a tool with potential in this field. Unfortunately, there are still few studies using this tool in older adults. The Dm variable could be used to quickly and easily identify those people who are in a situation of loss of physical performance, especially those whose condition does not allow them to perform functional tests (hospitalized people or those with limitations in ambulation).

Regarding the limitations of the study, it should be taken into account that both tensiomyography and myotonometry are tools for assessing the muscular quality of a muscle in isolation and without the participation of the central nervous system. Among the factors that influence physical performance, intermuscular coordination and the processing of information to conduct the task in question have a great influence on the final result of the functional test [44]. In this sense, it is normal that the correlation levels are moderate. As for the population, the recruitment of the sample included a relatively low percentage of subjects considered “frail”. Including a broader spectrum of the older adult population in future studies will allow a better understanding of these relationships.

## 5. Conclusions

The study found significant correlations between the SPPB test and the Dm values of the rectus femoris and vastus lateralis muscles. Additionally, there were significant correlations between the 5XSST test and the Dm values of the rectus femoris, as well as Dm and stiffness values in the vastus lateralis. However, no significant correlations were observed between tensiomyography and myotonometry measurements and the functional tests of TUG and walking speed. Similarly, there were no significant correlations between tensiomyography, myotonometry, handgrip strength, and isometric knee extension strength. Based on these findings, the authors suggest prioritizing functional tests for assessing older adults, at least until muscle quality assessment tools are further developed. However, further research on technologies that allow the evaluation of muscle quality is recommended.

## Figures and Tables

**Figure 1 healthcare-11-02169-f001:**
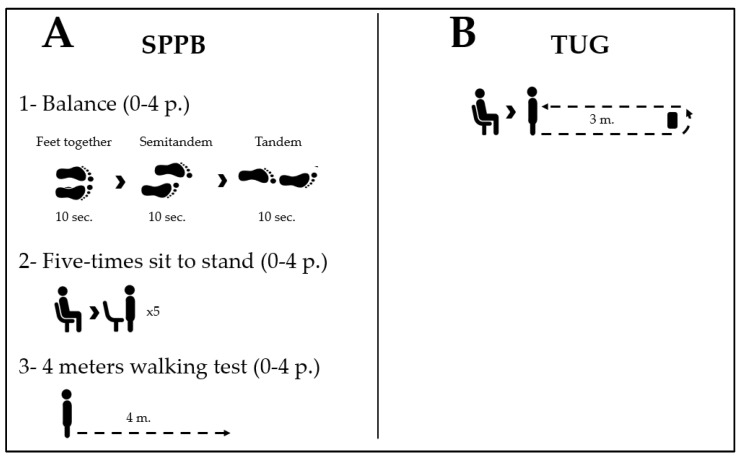
(**A**) Short Physical Performance Battery. (**B**) Timed Up and Go Test.

**Table 1 healthcare-11-02169-t001:** Anthropometric data of the sample.

Variable	Mean (SD) or N (%)
Sex	
Male	31 (60%)
Female	21 (40%)
Dominance	
Right	50 (96%)
Left	2 (4%)
Age	73.7 (7.44)
Height	159 (10.3)
Weight	67.4 (80.5; 60.1) *
BMI	28.3 (4.12)

*, median (Q1; Q3); BMI, body mass index; median (IQR); SD, standard deviation.

**Table 2 healthcare-11-02169-t002:** Descriptive analysis of the variables.

Variable	Mean (SD)	IQR
SPPB		12 (12; 11)
TUG		8.50 (10.1; 7.39)
5XSST		10.3 (11.9; 9.43)
4WT	1.07 (0.26)	
KE-MS	37.8 (15.8)	
HS		30.4 (39.9; 21.1)
Dm-RF	4.28 (2.26)	
Tc-RF	44.6 (14.7)	
St-RF	292 (45.1)	
Dm-VL		2.34 (3.42; 1.13)
Tc-VL		34.2 (58.2; 23.4)
St-VL	311 (36.7)	

IQR, interquartile range (median Q1; Q3); 5XSST, Five Times Sit to Stand Test; Dm, maximal radial muscle displacement; HG, handgrip strength; KE-MS; knee extension muscular strength; RF, rectus femoris; SD, standard deviation; St, stiffness; SPPB, Short Physical Performance Battery; Tc, contraction time; TUG, Timed Up and Go Test; VL, vastus lateralis; 4WT, 4 m Walking Test.

**Table 3 healthcare-11-02169-t003:** Correlation matrix.

		Rectus Femoris			Vastus Lateralis		
		Dm	Tc	St	Dm	Tc	St
**SPPB**	Correlation	0.491 †	0.137 †	0.073 †	0.329 †	0.113 †	0.273 †
	*p*-value	< 0.001	0.348	0.609	0.021	0.438	0.053
**TUG**	Correlation	−0.172 †	0.103 †	0.101 †	−0.067 †	0.032 †	0.040 †
	*p*-value	0.237	0.483	0.481	0.645	0.825	0.780
**5XSST**	Correlation	−0.340 †	−0.038 †	−0.108 †	−0.304 †	−0.076 †	−0.345 †
	*p*-value	0.016	0.793	0.447	0.032	0.602	0.012
**4WT**	Correlation	0.093 *	−0.076 *	−0.174 *	0.094 †	0.036 †	−0.098 *
	*p*-value	0.522	0.601	0.216	0.518	0.804	0.490
**HS**	Correlation	0.140 †	0.129 †	0.002 †	−0.047 †	−0.017 †	−0.092 †
	*p*-value	0.341	0.383	0.992	0.752	0.907	0.535
**KE** **-MS**	Correlation	0.220 *	−0.088 *	0.092 *	0.115 †	−0.085 †	0.108 *
	*p*-value	0.125	0.542	0.515	0.428	0.557	0.448

*, Pearson’s correlation coefficient; †, Spearman’s rank correlation coefficient; 5XSST, Five Times Sit to Stand Test; Dm, maximal radial muscle displacement; HS, handgrip strength; KE-MS; knee extension muscular strength; RF, rectus femoris; St, stiffness; SPPB, Short Physical Performance Battery; Tc, contraction time; TUG, Timed Up and Go test; VL, vastus lateralis; 4WT, 4 m Walking Test.

## Data Availability

Data sharing is not applicable to this article.

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
