# Peer review of "Correlation between Physical Performance and Tensiomyographic and Myotonometric Parameters in Older Adults"

_healthcare, 2023, doi:10.3390/healthcare11152169_

Round 1
Reviewer 1 Report
The present study investigates the correlation between the striated muscle strength parameters and the functional performance in healthy elderly. A group of 52 eldely people was subjected to the measurements of muscle strength parameters of rectus femoris and vastus lateralis, by myotonometry (stiffness) and tensiomyography (maximal radial muscle displacement,contraction time) along with several functional performance tests (SPPB, TUG, sit-to stand, walking speed and dynamometric tests).The results in a matrix form are subjected to the statistical analysis in order to identify possible correlations between measured parameters and functional performance. The authors have identified the positive correlation between radial displacement (r.f.) and stiffnes(v.l.) and SPPB test, concluding that functional tests still should be prioritized in older adults.
The study is well conceptualized and brings an interesting hypothesis, at least at the level of fundamental physiology. On the other hand, the Results and Methods sections should be more clearly written as they are a bit difficult to follow:
-In section 2.5 Variables, the muscle strength parameters should be clearly separated from functional tests. Equations should be written where applicable (stiffness). More complex functional tests require schemes or sketches.
- Names and abbreviations of variables described in 2.5 should match these in Table 3.
-All results are only represented through coefficients in Table 3. There should be some 2-dimensional plots representing correlations to corroborate the coefficients in Table 3.
-Handgrip strength test involves arm muscles while all other variables are related to the leg muscles. Could you briefly discuss?
Author Response
Revisor 1
The present study investigates the correlation between the striated muscle strength parameters and the functional performance in healthy elderly. A group of 52 eldely people was subjected to the measurements of muscle strength parameters of rectus femoris and vastus lateralis, by myotonometry (stiffness) and tensiomyography (maximal radial muscle displacement, contraction time) along with several functional performance tests (SPPB, TUG, sit-to stand, walking speed and dynamometric tests).The results in a matrix form are subjected to the statistical analysis in order to identify possible correlations between measured parameters and functional performance. The authors have identified the positive correlation between radial displacement (r.f.) and stiffnes (v.l.) and SPPB test, concluding that functional tests still should be prioritized in older adults.
The study is well conceptualized and brings an interesting hypothesis, at least at the level of fundamental physiology. On the other hand, the Results and Methods sections should be more clearly written as they are a bit difficult to follow:
Thank you very much for reviewing the article. We are convinced that our work will improve with your contributions.
-In section 2.5 Variables, the muscle strength parameters should be clearly separated from functional tests. Equations should be written where applicable (stiffness). More complex functional tests require schemes or sketches.
Thank you very much for your comment. The variables section has been subdivided according to whether they pertain to muscle quality, physical performance or muscle strength. In addition, the corresponding formulas have been added. Finally, two diagrams have been created in Figure format to explain the SPPB and TUG functional tests.
- Names and abbreviations of variables described in 2.5 should match these in Table 3.
Thank you for your input. We have proceeded to unify terms.
-All results are only represented through coefficients in Table 3. There should be some 2-dimensional plots representing correlations to corroborate the coefficients in Table 3.
Thank you very much for your comment. We have proceeded to include the correlation diagrams as supplementary material. As it was unfeasible to include all 36, we have proceeded to include only those 5 with statistical significance.
-Handgrip strength test involves arm muscles while all other variables are related to the leg muscles. Could you briefly discuss?
Thank you for your comment. Handgrip strength serves as a commonly employed indicator in both clinical and research domains for evaluating muscular strength. It is widely acknowledged as a reliable biomarker of overall health, and diminished grip strength values are indicative of potential occurrences such as frailty, sarcopenia, and other adverse events including falls, hospitalization, or mortality. While handgrip strength does not directly measure lower limb strength, numerous studies have established a significant correlation between handgrip strength and knee extension strength. In this sense, we have added the information in the introduction section, providing greater consistency in its subsequent evaluation.
Reviewer 2 Report
Overall this seems like a novel study with interesting findings.
Line 27- what variable of the rectus femoris and vastus lateralis?
Line 112-113, should be one sentence?
Line 116- same protocol...reword for clarity?
Line 118- also seems choppy- should this also be one sentence?
Line 121- Same comment
Line 125- same comment
Tables are rather confusing, could you display the IQR in a different column?
Table 3- consider making another heading for VL and RF with the subheadings so there is less looking down at the key.
Line 214, does Stiffness need to be capitalized?
Line 231- were there any other changes noted in this study mentioned
In your conclusion, can you make this more practitioner friendly? What did we learn about muscle quality?
Author Response
Revisor 2
Overall, this seems like a novel study with interesting findings.
Thank you very much.
Line 27- what variable of the rectus femoris and vastus lateralis?
Thank you for your input. We were referring to Dm, Tc (assessed by TMG) and stiffness (assessed by MMT). We have slightly modified the abstract to try to make it more understandable.
Line 112-113, should be one sentence?
Thank you. We have modified it.
Line 116- same protocol...reword for clarity?
Thank you for your input. We have added in the procedures section the entire protocol for taking measurements.
Line 118- also seems choppy- should this also be one sentence?
Thank you. We have modified it.
Line 121- Same comment
Thank you. We have modified it.
Line 125- same comment
Thank you. We have modified it.
Tables are rather confusing, could you display the IQR in a different column?
Thank you for your input. We have modified Table 2 by adding a column of our own for IQR.
Table 3- consider making another heading for VL and RF with the subheadings so there is less looking down at the key.
Thank you. We have modified the table to make it more understandable.
Line 214, does Stiffness need to be capitalized?
No. We have corrected it. Thank you very much.
Line 231- were there any other changes noted in this study mentioned
As the reviewer mentions, there were indeed more changes observed than in the Dm. An 18% increase was observed in the Tc of the gastrocnemius medialis. Meanwhile, the Dm increased by 24%, 26%, and 30% in the vastus medialis, biceps femoris, and gastrocnemius medialis muscles, respectively. The changes in Tc were not discussed in our manuscript, as the authors themselves acknowledge that the changes in Dm were greater and more consistent, across all the lower limb muscles studied. In contrast, the Tc exhibited a smaller change, limited to the gastrocnemius medialis muscle only.
In your conclusion, can you make this more practitioner friendly? What did we learn about muscle quality?
Thank you for your recommendation. We have rewritten the conclusions to make them more practical.
Reviewer 3 Report
Dear editor,
Thank you for the opportunity to evaluate the manuscript “Correlation between Physical Performance and Tensiomyographic and Myotonometric Parameters in Older Adults”. The objective of the present research was: To examine the correlation between physical performance and muscle strength and the variables obtained from tensiomyography and myotonometry. Below I send suggestions for the manuscript.
1 – The summaries are well written.
2 – The introduction is insufficient. The relationship between the analyzed variables is not clear. I notice that the innovative measurements (Tensiomyographic and Myotonometric) are measured in different conditions than the proposed tests. The authors need to present the relationship between the variables that will be measured. This is necessary, since the study is a correlation study.
3 – The authors must present the hypotheses after the objective. The hypotheses must have a theoretical foundation, presented throughout the introduction.
4 – The methods are well written
5 – The results are well written
6 – The discussion needs to be revised. I suggest that the authors present the results based on the presented hypotheses. Furthermore, it is necessary that the authors discuss the results found, within the specificity of the sample. In the discussion, the authors present the results of studies with young people and with athletes. These samples differ from the sample that was used in this manuscript.
7 – The conclusion was well written.
Although the manuscript is well written, I believe the authors need to make minor adjustments.
Author Response
Revisor 3
Thank you for the opportunity to evaluate the manuscript “Correlation between Physical Performance and Tensiomyographic and Myotonometric Parameters in Older Adults”. The objective of the present research was: To examine the correlation between physical performance and muscle strength and the variables obtained from tensiomyography and myotonometry. Below I send suggestions for the manuscript.
We appreciate your feedback and are confident that with your input the study will have improved.
1 – The summaries are well written.
Thank you.
2 – The introduction is insufficient. The relationship between the analyzed variables is not clear. I notice that the innovative measurements (Tensiomyographic and Myotonometric) are measured in different conditions than the proposed tests. The authors need to present the relationship between the variables that will be measured. This is necessary, since the study is a correlation study.
Thank you for your recommendation. the introduction has been expanded and restructured to make it more relevant to the purpose of the study.
3 – The authors must present the hypotheses after the objective. The hypotheses must have a theoretical foundation, presented throughout the introduction.
Thank you for your input. We have added the main hypothesis of the article.
4 – The methods are well written
Thank you.
5 – The results are well written
Thank you.
6 – The discussion needs to be revised. I suggest that the authors present the results based on the presented hypotheses. Furthermore, it is necessary that the authors discuss the results found, within the specificity of the sample. In the discussion, the authors present the results of studies with young people and with athletes. These samples differ from the sample that was used in this manuscript.
Thank you very much for the recommendation. We have proceeded to add data from recent studies related to TMG in older adults in the discussion.
7 – The conclusion was well written.
Thank you.
Although the manuscript is well written, I believe the authors need to make minor adjustments.
Thank you very much. We have revied it in order to correct the mistakes.
Round 2
Reviewer 1 Report
The authors have properly addressed my questions. I consider that the manuscript is acceptable for publication.
Author Response
Thank you very much